# Screening and Isolation of Xylanolytic Filamentous Fungi from the Gut of Scarabaeidae Dung Beetles and Dung Beetle Larvae

**DOI:** 10.3390/microorganisms12030445

**Published:** 2024-02-22

**Authors:** Livhuwani Makulana, Daniel C. La Grange, Kgabo L. M. Moganedi, Marlin J. Mert, Nkateko N. Phasha, Elbert L. Jansen van Rensburg

**Affiliations:** Department of Biochemistry, Microbiology and Biotechnology, University of Limpopo, Limpopo Provice, Private Bag X1106, Sovenga 0727, South Africa; danie.lagrange@nwu.ac.za (D.C.L.G.); kgabo.moganedi@ul.ac.za (K.L.M.M.); marlin.mert@ul.ac.za (M.J.M.); nkateko.phasha@ul.ac.za (N.N.P.); elbert.jansenvanrensburg@ul.ac.za (E.L.J.v.R.)

**Keywords:** Scarabaeidae, xylanase activity, gut-inhabiting fungi, biofuel, phylogeny

## Abstract

Research on renewable biotechnology for renewable biofuel applications has reached new heights. This is highlighted by extensive biomining for novel enzymes to reduce the production costs from animal and insect gut microbiomes. This study explored the diversity and composition of hemicellulolytic fungi in the gut microbiota from dung beetles of the family Scarabaeidae (*Pachylomerus femoralis*, *Anachalcos convexus* and *Euoniticellus intermedius*). Two hundred and twenty-two filamentous fungi were isolated, purified and identified using rDNA sequencing of the ITS and D1/D2 regions. The fungal isolates were assigned to 12 genera and 25 species. Fungi associated with the genus *Aspergillus* was in abundance, with *Hypocrea lixii* predominantly isolated. Isolates that produced more than 3 U/mL of xylanase activity were evaluated further. The highest xylanase activity was of 23.6 and 23.5 U/mL for L1XYL9 (*E. intermedius* larvae) and *Hypocrea lixii* AB2A3 (*A. convexus*), respectively. Phylogeny of the fungal strains with xylanolytic activity was analysed using ITS rDNA sequences and revealed close genetic relatedness between isolates from the different dung beetle species. Fungal genera commonly found in the gut of both adult beetles and larvae included *Aspergillus*, *Hypocrea, Talaromyces and Penicillium*. The results obtained in this study suggest that the gut of Scarabaeidae dung beetles in South Africa is a rich source of xylanolytic fungi.

## 1. Introduction 

Enzymes are preferred as the catalysts for many industrial applications, as chemical catalysts typically require extreme temperatures and pressure and these often have a negative impact on the environment [1]. Enzymes have recently gained attention for the breakdown of plant biomass into fermentable sugars for the production of products such as biofuels, animal feeds and chemicals [2]. Enzymes such as xylanases have received attention for their application in various industries such as animal feed, biofuel, baking, pulp and paper, liquefaction of fruits and vegetables, clarification of beer and juices, as well as bioremediation [3,4]. In the biofuel industry, xylanases are used for the efficient hydrolysis of hemicelluloses. The addition of xylanase during enzymatic hydrolysis of lignocellulose polymers increased the rate and the efficiency of the process [3,5], since xylanases help expose the cellulose to cellulase attack [6]. Enzymatic hydrolysis is a sustainable, effective and environmentally friendly method. This process entails breaking down glycosidic bonds within the polymers present in lignocellulosic materials to produce fermentable sugars. To achieve successful and efficient enzymatic hydrolysis of lignocellulosic plant biomass, xylanase enzymes play a pivotal role. Xylanases have been previously isolated from a diverse array of naturally occurring life forms including plants, animals, insects and microorganisms [7].

The symbiotic relationship between microorganisms and insects has been a subject of extensive study, aiming to gain insights into their evolutionary diversification. Fungal mutualism with insects holds significant importance in the developmental stages of insects and their overall fitness. This is achieved through the provision of nitrogen compounds, the degradation of high molecular weight molecules, and the production of pheromones for mating and communication [8]. Recently, these microorganisms have received attention for their lignocellulolytic activity, presenting promising applications in biofuel production [4,8]. Despite the widespread study of microorganisms associated with insects, their isolation from the gut of dung beetles remains limited. The gut microbiota of adult dung beetles and their larvae represent a unique source of xylanases. Dung beetles are a major group of insects (Order Coleoptera; Family Scarabaeidae) with approximately 370,000 species identified worldwide, of which 8000 of those species have been reported to be found in the Kruger National Park in South Africa [8,9]. Dung beetles are known to feed on wet and dry dung of herbivores which consists of 80% indigestible material such as cellulose, hemicellulose, lignin, tannin, chitin and other waste materials. This material is degraded by lignocellulolytic enzymes (xylanase and cellulase) produced by microorganisms in the digestive tract of the dung beetles [4]. These microorganisms further assist the digestion system of dung beetles by performing essential functions such as nutrient production and compound detoxification [10]. 

Currently, most research on gut microbiomes is mainly focused on termites, honeybees and mosquitoes [4,11,12]. The available information on the diversity and composition of the gut microbiota in dung beetles, particularly focusing on hemicellulolytic microbes, is currently limited. This knowledge gap extends to the fungal microbiota associated with these beetles [4,11]. Fungi are well known for their remarkable ability to produce extracellular enzymes important in biotechnological applications. Fungal species such as *Aspergillus* and *Trichoderma* are well-known for their ability to produce xylanase enzymes that are used commercially. Other fungal species include *Talaromyces* [5] and *Rasamsonia emersonii* [13]. In this study, the guts of two adult dung beetles of the Family Scarabaeidae (*Pachylomerus femoralis* and *Anachalcos convexus*) and gut of one dung beetle larva (*Euoniticellus intermedius*) (Figure 1) from the same family were screened for fungi with xylanolytic activity.

## 2. Materials and Methods

### 2.1. Collection and Dissection of Dung Beetles

Dung beetle larva (*Euoniticellus intermedius*) and dung beetles (*Pachylomerus femoralis* and *Anachalcos convexus*) were donated by Prof Byrne, School of Animal, Plant and Environmental Sciences at the University of the Witwatersrand, South Africa. The dung beetles were maintained and anaesthetised following the method described in [15]. The beetles were surface sterilised by washing them in 70% ethanol for 10 min and subsequently washed with sterile distilled water before dissection. The dung beetles or dung beetle larvae were individually placed on a flame-sterilised glass slide and sterile forceps were used to remove the elytra, which allowed easy access to the gut content [12]. The entire gut of the dung beetles or dung beetle larva was homogenised by suspending the content in 10 mL of a 0.7% saline solution. 

### 2.2. Isolation of Filamentous Fungi from the Gut of Dung Beetles/Larvae

One millilitre of the homogenised dung beetle or larva gut was plated on 0.67% YNB (Yeast Nitrogen Base, Difco) with amino acids containing 2% xylan as a carbon source and 0.2% chloramphenicol to inhibit bacterial growth [16]. Plates were incubated at 30 °C for up to 5 d. Different morphotypes were purified several times by placing 10 × 5 mm agar block repeatedly on YM plates (10 g/L glucose, 0.2 g/L chloramphenicol, 3 g/L malt extract, 3 g/L yeasts extract, 5 g/L peptone and 15 g/L bacteriological agar) [17]. The process was repeated until pure colonies were obtained. The purified fungal isolates were inoculated on YM plates and incubated at 30 °C for 5 d. Following incubation, 1 mL of sterile distilled water was added and spores were dislodged with a glass rod. The spore suspension was added to 1 mL 30% glycerol solution and stored at −80 °C until use.

### 2.3. ITS and D1/D2 Sequencing

All fungal isolates were sent to Inqaba Biotechnical Industries (Pty) Ltd., South Africa for sequencing of the ITS and D1/D2 rDNA regions. DNA was extracted using the ZR Fungal DNA MiniPrepTM Kit (Zymo Research) according to the manufacturer’s instructions. The ITS1-5.8S-ITS2 region was amplified using PCR primers ITS-1 (5′-TCC GTA GGT GAA CCT GCG G-3′) and ITS-4 (5′-TCC TCC GCT TAT TGA TAT GC-3′). Amplification was carried out in 25 µL reactions using the EconoTaq Plus Green Master Mix (Lucingen). The following PCR conditions were used: 35 cycles including an initial denaturation at 95 °C for 2 min. Subsequent denaturation at 95 °C for 30 s, annealing at 50 °C for 30 s and extension at 72 °C for 1 min. A final extension at 72 °C for 10 min was followed by holding at 4 °C. Additionally, the D1/D2 domain of the 26S rDNA region was amplified using primers NL1 (5‘-GCA TAT CAA TAA GCG GAG GAA AAG-3′) and NL4 (5′-GGT CCG TGT TTC AAG ACG G-3′) as described above. DNA sequencing was performed using ABI V3.1 BigDye according to the manufacturer’s instructions on an ABI 3500 XL Instrument [17]. 

### 2.4. Identification of Fungal Isolates and Phylogenetic Analysis

The ITS and D1/D2 sequencing data obtained were cleaned, trimmed and aligned using Bio-edit software version 7.0, http://www.mbio.ncsu.edu/bioedit/page2.html, accessed on 10 January 2020). Sequence alignment was performed using the Muscle software package as implemented in the MEGA 7 program [18]. Furthermore, the ITS and D1/D2 sequences were compared to sequences in the Genbank database using the basic local alignment search tool (BLAST) of the National Center for Biotechnology Information (http://www.ncbi.nlm.nih.gov/, accessed on 10 March 2020). A phylogenetic tree was constructed based on the ITS domains using MEGA version 7.0 [18]. The evolutionary history was inferred using the neighbor-joining method [19]. Bootstrap analysis [20] was performed from 1000 replications to determine the confidence levels of clades, and only values >50% were recorded on the phylogenetic tree [19].

### 2.5. Screening Filamentous Fungal Isolates for Xylanase Activity 

A 15 mm × 2 mm block of agar was cut from a 5-d-old fungal culture plate and inoculated into test tubes containing 5 mL 0.67% YNB with amino acids and 1% beechwood xylan as the carbon source. The test tubes were incubated on a rotary shaker at 30 °C for 72 h. After incubation, quantitative analysis for xylanase activity was performed using the DNS method [20]. The xylanase enzyme activity was expressed in katals per millilitre (U/mL), where one unit of xylanase is defined as the amount of enzyme that liberates 1 µmol of xylose equivalents per minute. 

## 3. Results

In this study, the guts of three Scarabaeidae dung beetles (*Pachylomerus femoralis and Anachalcos convexus*) and one dung beetle larva (*Euoniticellus intermedius)* were isolated, identified and screened for xylanolytic fungi. A total of 222 filamentous fungi were isolated from *P. femoralis*, *A. convexus* and *E. intermedius*. A total of two hundred and seventeen xylanase-producing filamentous fungi were isolated from the guts of *P. femoralis* (35 fungal isolates), *A. convexus* (118 fungal isolates) and the larvaof the dung beetle *E. intermedius* (69 fungal isolates). The phylogenetic tree for all fungal isolates with xylanolytic activity was deduced to evaluate their relationship. 

### 3.1. Identification and Characterisation of Fungal Isolates

All fungal isolates were identified and characterised by sequencing the ITS and D1/D2 regions. The results revealed that the isolates belonged to ten different genera, namely *Aspergillus, Hypocrea* (previously *Trichoderma*)*, Penicillium*, *Mucor*, *Neosartorya*, *Rhizopus, Talaromyces*, *Taifanglania and Byssochlamys.* The most prevalent genus was *Aspergillus* (89 strains) followed by *Hypocrea* (81 strains), *Penicillium* (28 strains) and *Neosartorya* (11 strains), while the less dominant isolates belonged to *Talaromyces* (7 strains) and *Rhizopus* (3 strains) with *Byssochlamys, Taifanglania* and *Emericella* represented by 1 strain each (Figure 2A–C).

The gut of *A. convexus* yielded 118 strains, of which *Aspergillus* was the most abundant followed by *Hypocrea* and *Penicillium* (Figure 2A). The gut of dung beetle larva yielded 64 strains represented by 10 different genera. The most dominant genus in the larva gut was *Hypocrea*, followed by *Aspergillus* and *Neosartorya* (Figure 2B). Thirty-five isolates were obtained from the gut of *P. femoralis*, of which *Aspergillus* was also the dominant genus, followed by *Trichoderma* and *Penicillium* (Figure 2C). 

### 3.2. Xylanase Production by Fungal Isolates

All isolates obtained in this study were screened for xylanase activity using xylan as a substrate. Two hundred and seventeen were xylanolytic, with 203 isolates showing xylanase activity higher than 3 U/mL (Figure 3, Figure 4 and Figure 5). All isolates with more than 3 U/mL were -evaluated further. Xylanolytic isolates belonged to 10 genera, namely *Aspergillus* (78), *Hypocrea* (77), *Penicillium* (27), *Talaromyces* (4), *Neosartorya* (6), *Mucor* (4), *Geotrichum* (3), *Rhizopus* (2), *Byssochlamys* and *Taifanglania* (1). 

The dung beetle *A. convexus* yielded the most fungi, with xylanase activity higher than 3 U/mL (Figure 3). The best producers were *Hypocrea lixii* AB6XYL2 and AB2A3 with 24 and 19 U/mL, respectively. This was followed by *Aspergillus* sp. AB2XYL30 (15 U/mL), *Penicillium janthinellum* AB1XYL14 (15 U/mL) and *Hypocrea* sp. AB1XYL4 (12 U/mL). 

*Hypocrea* sp. (BB2A1 and BB1A3) from the gut of *P. femoralis* (Figure 4) both produced the highest xylanase activity of 12 U/mL. This was followed by *A. niger* BB2XYL21 (10 U/mL) and *T. helicus* BB2X4 (9 U/mL). 

Thirty-five isolates from *E. intermedius* showed xylanase activity higher than 3 U/mL (Figure 5). *H. lixii* was the most dominant species, with its xylanase activity ranging from 4–13 U/mL. *H. lixii* is the teleomorph of *Trichoderma harzianum*. *H. lixii* was an abundant species in the gut of both adult and dung beetle larva. *A. fumigatus* L1XYL9 produced the highest xylanase activity of 24 U/mL followed by *B. spectabilis* L2A4 producing 19 U/mL. 

Thirteen fungal isolates that showed higher xylanolytic activity as compared to other species were selected for further study (Figure 6). Some species, such as Rhizopus microsporus (AB4A5), were not selected because of their higher xylanolytic activity but because they were unique gut microbiome species.

### 3.3. Phylogenetic Analysis 

The phylogenetic trees were prepared using ITS sequencing data for the individual isolates (Figure 7, Figure 8, Figure 9 and Figure 10). Three phylogenetic trees were constructed, representing the species with xylanolytic activity from the gut of *A. convexus*, *P. femoralis* and *E. intermedius*. 

The species isolated from the gut of *A. convexus* represent several clades, namely *Trichocomaceae, Penicillium*, *Aspergillus* and *Trichoderma* (Figure 7). The family *Trichocomaceae* split into three separate families, which were *Aspergillaceae*, *Trichocomaceae* and *Thermoascaceae*. This is reflected in the phylogenetic data of the *Trichocomaceae* clade I, composed of *Talaromyces, Aspergillus* and *Penicillium* species. The *Mucoraceae* clade was represented by *Hyphomucor, Mucor* and *Rhizopus* genera. The relationship among the species was well supported by branches > 50%. 

*P. femoralis* represented two clades, namely *Trichocomaceae* and *Trichoderma* (Figure 8). Phylogeny of *E. intermedius* isolates deduced three monophyly groups of *Penicillium*. Other clades were *Neosartorya, Aspergillaceae, Talaromyces*, *Trichoderma* and *Byssochlamyces* (Figure 9). These clades, besides *Hypocrea,* are part of the *Trichocomaceace* family. The other clade deduced was the *Mucoraceae* clade made up of *Mucor circinelloides* and *Rhizopus* sp.

All the fungal isolates with xylanolytic activity from the gut of *A. convexus*, *P. femoralis* and *E. intermedius* were used to construct a best-scoring neighbor-joining tree to evaluate the relatedness of their fungal species. Fungal species such as *H. lixii, Hypocrea* sp., *A. niger*, *Neosartorya fischeri* and *Talaromyces helicus* from the gut of *E. intermedius* showed relatedness as they formed clades with fungal species from the gut of *A. convexus* and *P. femoralis*. The deduced branches were well-supported with a bootstrap of > 50%. The deduced phylogenetic tree showed that the fungal species isolated from *A. convexus, P. femoralis* and *E*. *intermedius* are closely related since most species formed clades with bootstrap values > 50% (Figure 10). 

## 4. Discussion

Lignocellulose is the predominant component of woody plant material, soil organic material and decaying wood, making it the most abundant form of biomass in terrestrial ecosystems. Lignocellulose is a cheap, abundant, renewable substrate that can be used for the production of biofuel. Lignocellulose is composed of cellulose and hemicellulose polymers that can be broken down into their monomers, and these can be fermented to ethanol. Currently, several constraints are affecting the use of biomass for biofuel. One of these constraints is finding cost-efficient enzymes that degrade lignocellulose polymers into monomers. The degradation efficiency could be improved with the inclusion of the endo-xylanase enzyme. The gut of the dung beetle is a rich source of hemicellulolytic microorganisms [4,12]. In this study, the guts of two Scarabaeidae dung beetles (*A. convexus* and *P. femoralis*) and one dung beetle larva (*E. intermedius)* were screened for xylanolytic fungi. A total of 222 filamentous fungi were isolated from *A. convexus*, *P. femoralis* and *E. intermedius*. Two-hundred and seventeen were found to have xylanase activity.

The abundance of genera such as *Aspergillus*, *Hypocrea*, *Penicillium* and *Neosartorya* in the guts of *A. convexus* and *P. femoralis* and one dung beetle larva (*E. intermedius)* indicates that the gut of dung beetles is a good source for the isolation of xylanolytic microorganisms. These genera are one of the well-known cellulase and xylanase producers, with *Hypocrea jecorina* (previously *Trichoderma reesei*) and *Aspergillus niger* being the most studied species and the species widely used for the production of a commercial enzyme [16,21,22]. The species isolated from the adult dung beetles (*A. convexus* and *P. femoralis*) were similar but differed from those isolated from the dung beetle larva (*E. intermedius)*. The difference in species composition between the adult beetles and the larva could be caused by the moulting process that occurs several times during the development stage. During this process, the larva shed not only the hind- and foregut lining but also the inhabiting microorganisms [11]. In addition, species composition is expected to differ between different developmental stages [10]. According to [10], the difference in species composition could also be due to the difference in diet [11], the environment [23], and social interactions [24].

Most filamentous fungal isolates had xylanolytic activity and only a few did not exhibit any xylanolytic activity. Filamentous fungi are well-known for their ability to produce xylanase enzymes and are widely used in industry for enzyme production, especially species such as *Hypocrea jecorina* and *Aspergillus niger* [25,26]. The fungal isolates that showed the highest xylanase activity were *A. fumigatus* L1XYL9 isolated from dung beetle larva (*E.* intermedius) with xylanase activity of 24 U/mL. The xylanase activity produced by *A. fumigatus* L1XYL9 in this study was higher than that by *A. fumigatus* GGV – BT 03 (0,1 U/mL) reported by [27], *A. fumigatus* Z5 (15 U/mL) reported by [28] and *A. Fumigatus* MA28 (9 U/mL) reported by [29]. This was the same as xylanase activity exhibited by *H. lixii* AB6XYL2 isolated from the gut of the adult dung beetle *A. convexus*. The xylanase activity produced by *H. lixii* AB6XYL2 in this study was higher than that produced by *H. lixii* reported by [30], with 24 U/mL of xylanase activity. *Hypocrea* species are well-known hyper-producers of industrially important enzymes including lignocellulolytic enzymes. *Hypocrea and Aspergillus* species are well-known for their ability to degrade polymers (cellulose and hemicellulose) [25,26]. The xylanase activity of these gut microbiota is notably lower compared to the commercial xylanase [31]. This disparity arises from the absence of an optimization strategy, a crucial factor that could substantially elevate xylanase activity. Implementing a targeted optimization approach is essential to unlock the full potential of the gut microbiota’s xylanase capabilities, thereby bridging the performance gap with commercial counterparts. The dominance of these fungal species in the gut of these dung beetles revealed that the gut of dung beetles/larvae is a good source for the isolation of fungal species that will play an important role in the biofuel industry. 

The dominant fungal species in the guts of the adult dung beetles was *A. niger*. Its presence in the gut of dung beetles was in agreement with the study by [32] which found *A. niger* to dominate the gut of the three insect species collected from Assiut Governorate. Most of the *A niger* isolates in this study were able to produce xylanases. The co-operative action between *A. niger* and *H. lixii* in degrading polymers such as cellulose and hemicellulose reflects a symbiotic association in the gut microbiomes of dung beetles. Given the variations in composition, determining whether it is more beneficial to isolate the microbiota from the larvae or the gut becomes challenging. Further species isolated in this study included species such as *A. terreus* [32], *R. oryzae* [33], *P. echinulatum* [34] and *T. helices* [35] that have been used in the biofuel industry. In this study, these isolates showed good xylanase activity. The deduced phylogenetic tree reflected a close relationship between the fungi with xylanolytic activity from the gut of the two adult dung beetle and the larvae species. This further shows that the species that showed lower xylanolytic activity could be enhanced through evolutionary adaptation. Evolutionary adaptation has been extensively employed for strain improvement in industrial applications. Fungal strains exhibiting lower xylanase activity in this study can be enhanced through evolutionary adaptation, following an initial optimization strategy aimed at increasing xylanase production [36]. The fungal species such as *Taifanglania* sp. AB1XYL6 isolated in this study did not show any xylanase activity, but did, however, form a paraphyletic group with the genus *Acrophialophora,* which is a thermotolerant soil fungus that is widely distributed in temperate and tropical regions. *Acrophialophora nainiana* has been reported for its ability to produce cellulase and xylanase enzymes [37]. Since it shares a common recent ancestor with *Taifanglania* sp., they may be paraphyletic because *Taifanglania* sp, lost some of its characteristics (lignocellulolytic activity) that it shared with *Acrophialophora*. The genus *Taifanglania* further forms a monophyletic group with the genus *Chaetomium*. The genus *Chaetomium* is known for its ability to colonise different substrates. Most species from this genus are capable of producing cellulase that degrades cellulose, resulting in the production of different bioactive metabolites [38]. 

It is very difficult to be certain if the isolated species are endosymbionts of the gut of the Scarabaeidae dung beetle or if this species will differ from one Scarabaeidae species to the other since the microbial species could be transitory inhabitants associated with host feeding habits or might even be using the insect as a dispersal mechanism [13]. 

In conclusion, the results of this study show that the filamentous fungi associated with the guts of dung beetle/larvae are highly diverse in terms of the number of species, while phylogenetic analysis showed their close relatedness. The findings are in agreement with the report that the gut of the dung beetle is a rich source of hemicellulolytic microorganisms. The gut habitats have consortia that are acting synergistically to provide many of the nutritional needs of the beetle host. In addition, the degradation of lignocellulosic materials is reflected by the high percentage of filamentous fungi with xylanolytic ability. This is the first study to report the distribution of xylanase-producing fungal species inhabiting the gut of the *A. convexus*, *P. femoralis* and *E. intermedius* larvae. This study recommends that the highest xylanase-producing fungi should be investigated further to improve the xylanase activity and be screened for other enzymes related to lignocellulose degradation. These fungal isolates could be ideal in the hydrolysis of cheap lignocellulose substrates, such as grasses.

## Figures and Tables

**Figure 1 microorganisms-12-00445-f001:**
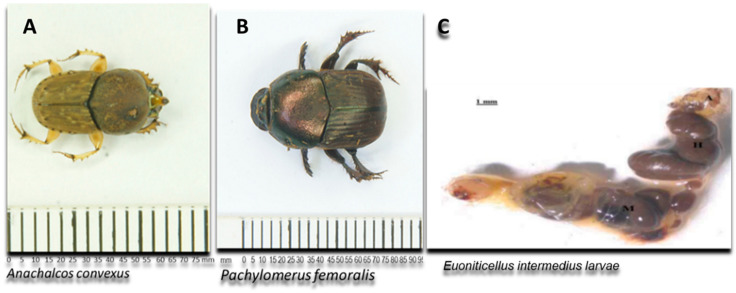
Schematic representation of the adult dung beetles *Pachylomerus femoralis* and *Anachalcos convexus* and the gut of dung beetle larva *Euoniticellus intermedius* [14] used in this study.

**Figure 2 microorganisms-12-00445-f002:**
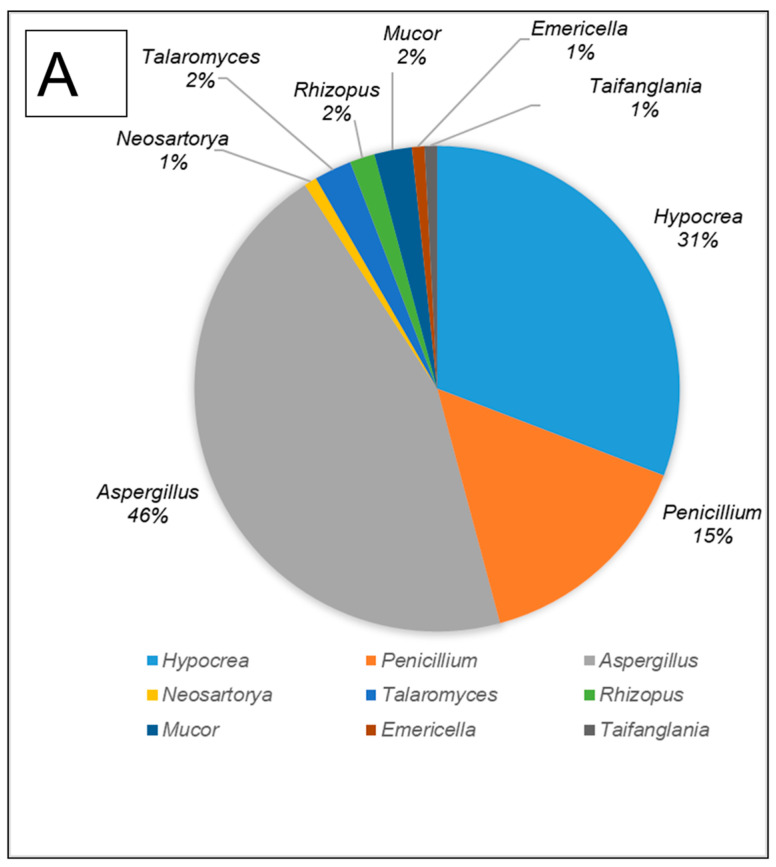
The filamentous fungi isolated from the gut of dung beetles and dung beetle larva. (**A**) gut microbiome from *A. convexus* (**B**) gut microbiome from *E. intermedius* larva and (**C**) gut microbiome from *P. femoralis*.

**Figure 3 microorganisms-12-00445-f003:**
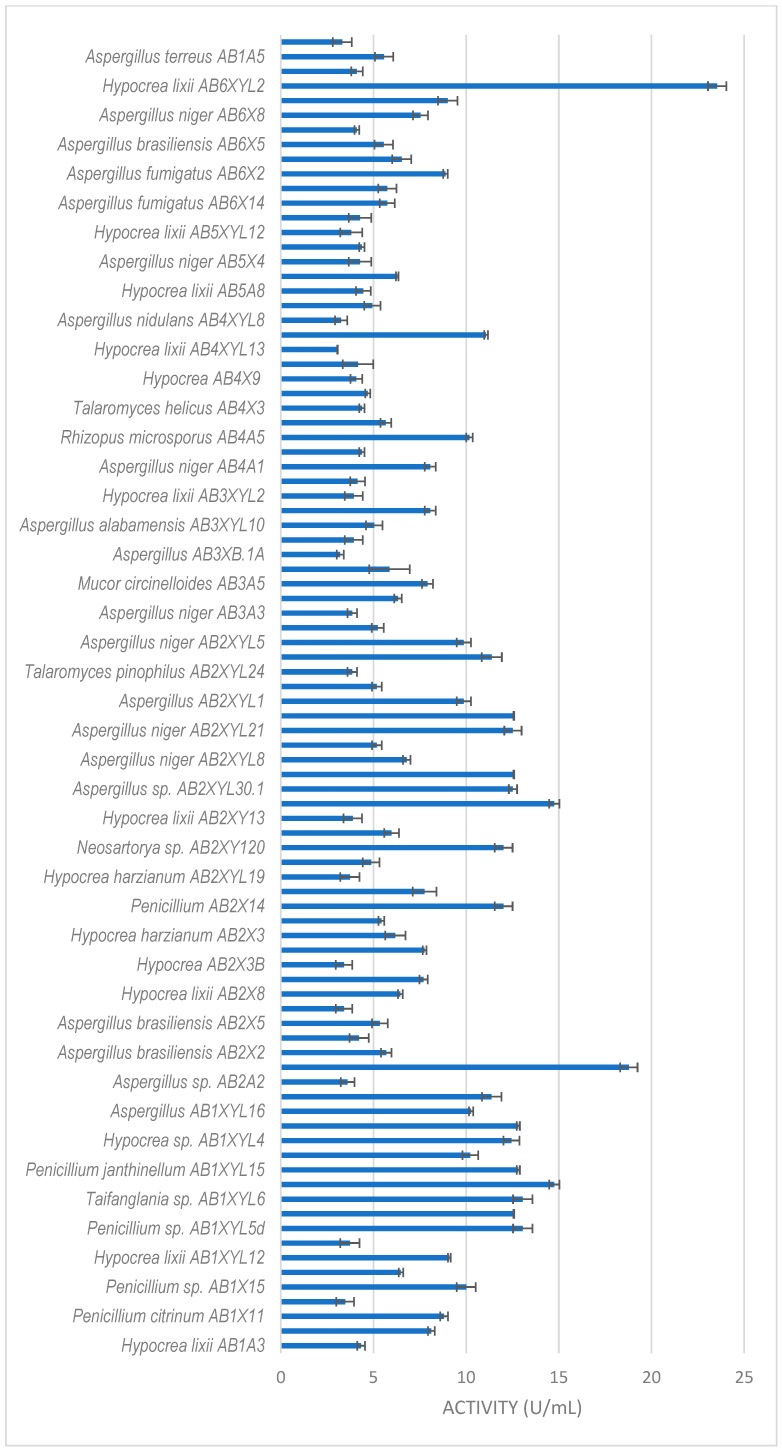
Xylanase activity by filamentous fungi isolated from the gut of the dung beetle *A. convexus.* The results are the mean of three independent experiments with standard deviation values.

**Figure 4 microorganisms-12-00445-f004:**
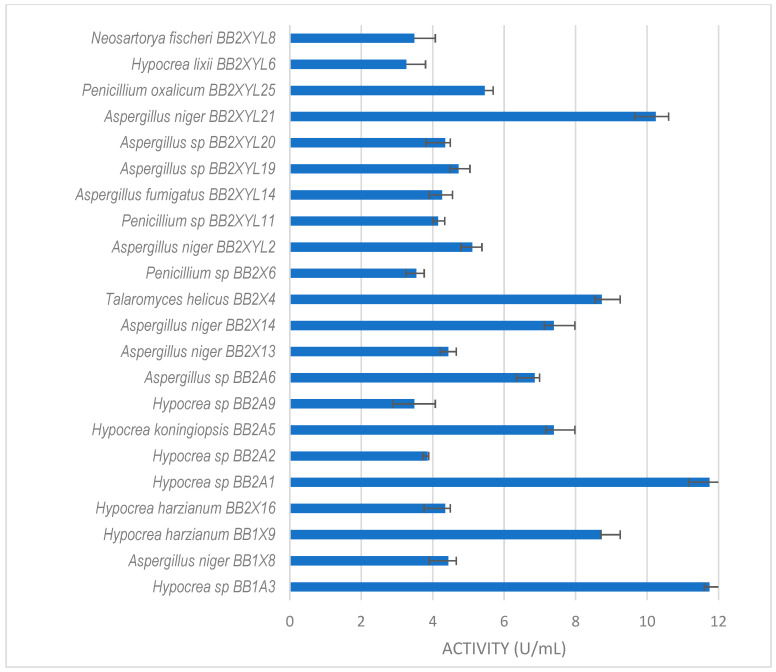
Xylanase activity by filamentous fungi isolated from the gut of the dung beetle *P. femoralis.* The results are the mean of three independent experiments with standard deviation values.

**Figure 5 microorganisms-12-00445-f005:**
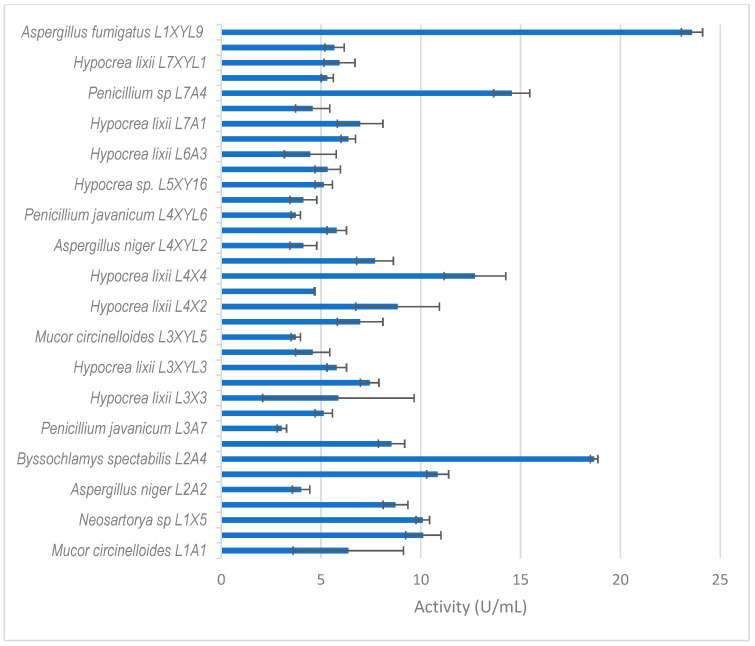
Xylanase activity by filamentous fungi isolated from the dung beetle *E. intermedius* larva. The results are the mean of three independent experiments with standard deviation values.

**Figure 6 microorganisms-12-00445-f006:**
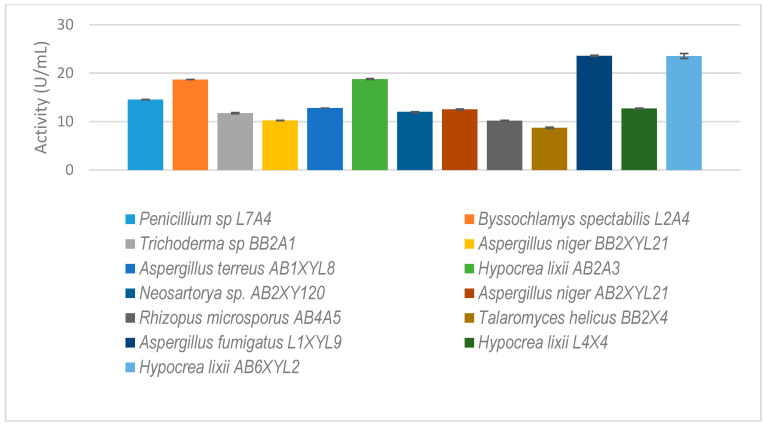
The isolated gut-inhabiting fungi from *P. femoralis* and *A. convexus* and dung beetle larva *E*. *intermedius* selected for further study. The results are the mean of three independent experiments with standard deviation values.

**Figure 7 microorganisms-12-00445-f007:**
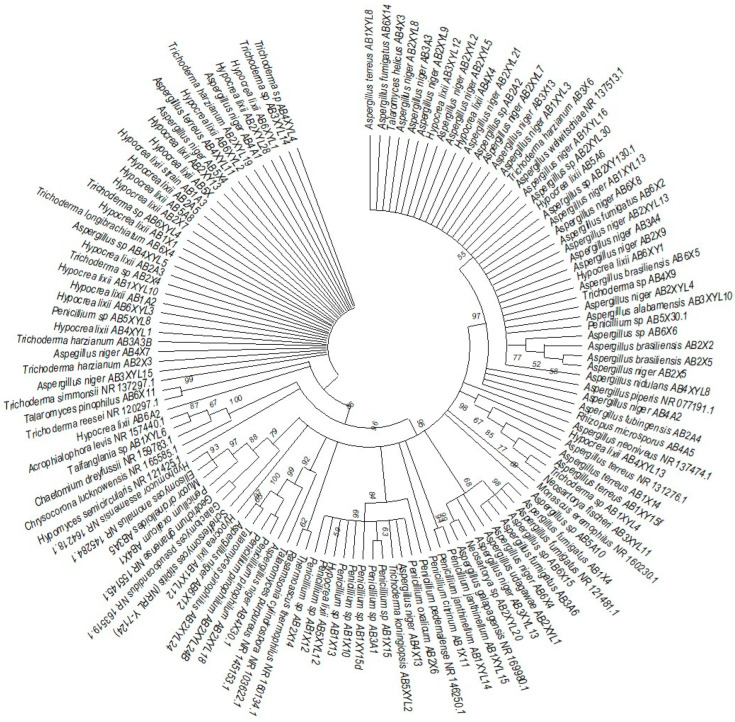
Neighbor-joining tree deduced using an ITS sequence of different strains isolated from the gut of the dung beetle *A. convexus*. Only branches with more than 50% bootstrap support are shown. *S. stipitis* NRRL-Y-7124 was used as an outgroup.

**Figure 8 microorganisms-12-00445-f008:**
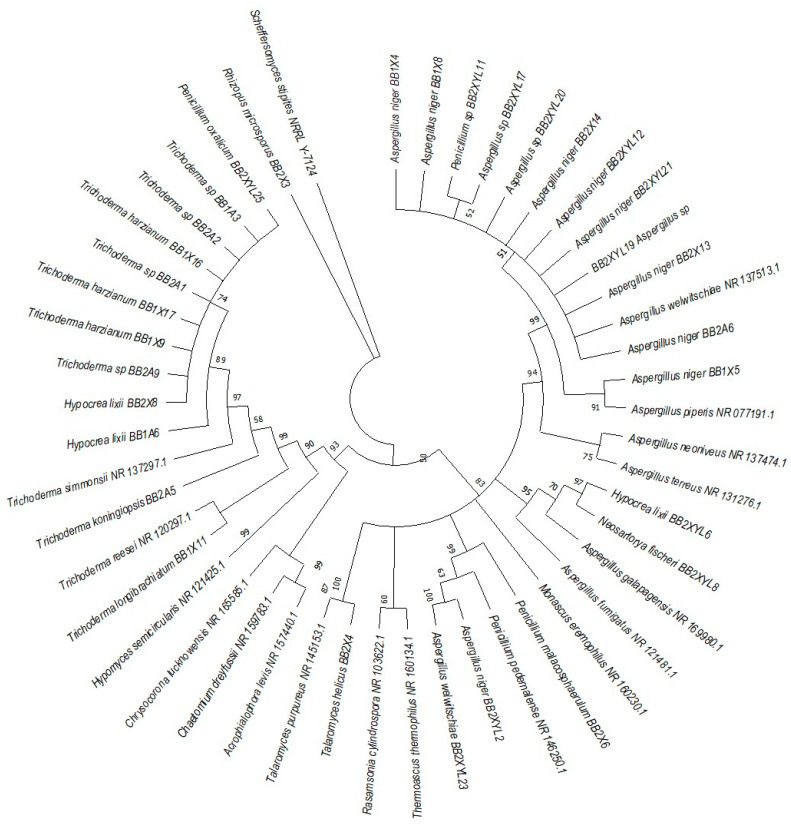
Neighbor-joining tree was deduced using an ITS sequence of different strains isolated from the gut of the dung beetle *P. femoralis*. Only branches with more than 50% bootstrap support are shown. *S. stipitis* NRRL-Y-7124 was used as an outgroup.

**Figure 9 microorganisms-12-00445-f009:**
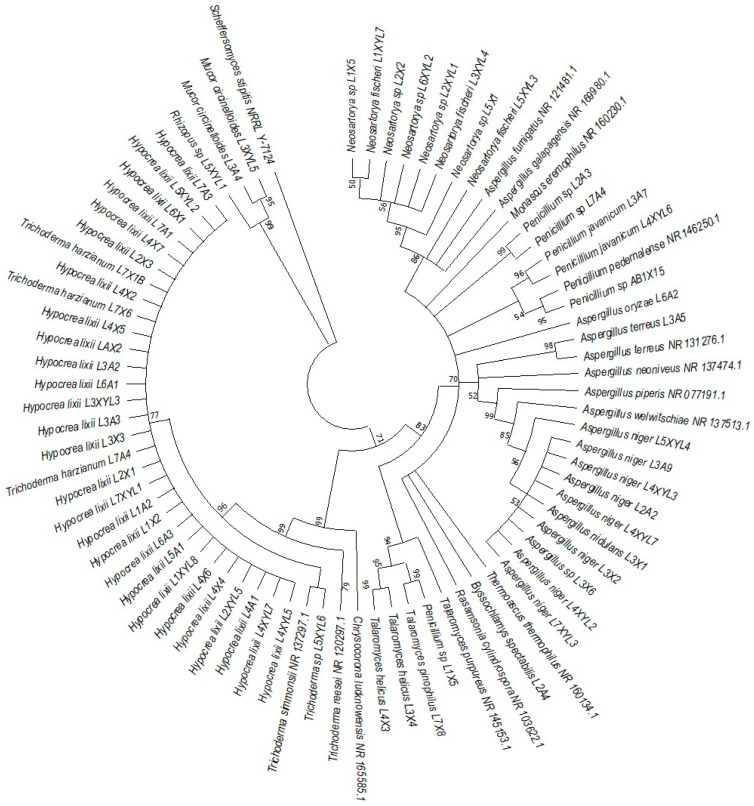
Neighbor-joining tree deduced using the ITS sequence of different strains isolated from the gut of the dung beetle larva *E. intermedius*. Only branches with more than 50% bootstrap support are shown. *S. stipitis* NRRL-Y-7124 was used as an outgroup.

**Figure 10 microorganisms-12-00445-f010:**
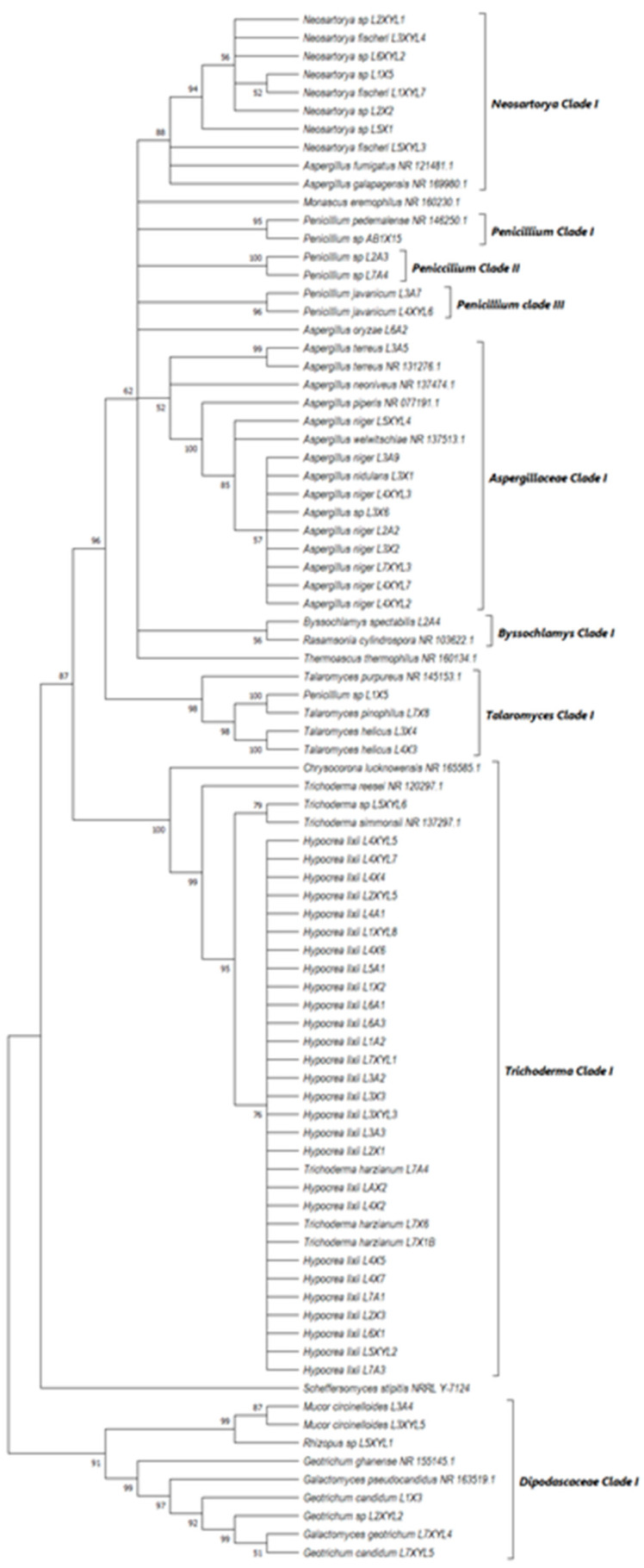
Neighbor-joining tree deduced using the ITS sequence of fungal strains isolated from the guts of the dung beetles *P. femoralis* and *A. convexus* and the dung beetle larva *E*. *intermedius*. Only branches with more than 50% bootstrap support are shown. *S. stipitis* NRRL-Y-7124 was used as an outgroup.

## Data Availability

The original contributions presented in the study are included in the article, further inquiries can be directed to the corresponding author.

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
