# Peer review of "Screening and Isolation of Xylanolytic Filamentous Fungi from the Gut of Scarabaeidae Dung Beetles and Dung Beetle Larvae"

_microorganisms, 2024, doi:10.3390/microorganisms12030445_

Round 1

Reviewer 1 Report

Comments and Suggestions for Authors

It is a very interesting paper, well written and well explained that deserves to be published after minor corrections.

- Line 83, 85, 92, lines 162 – 173, and all over the manuscript – although both symbols (l or L) can be used for “liter”, please standardize the nomenclature by using, e.g. “L”.

- line 132 – Trichoderma is the old name of Hypocrea genera? If so, why are they separate in disc diagrams of Figure 3? Explain the reason of using both names.

- lines 143 and 144 – This information corresponds to Figure 3 C (and not Figure 3B!)

- lines 145 and 146 - This information corresponds to Figure 3 B (and not Figure 3C!)

- line 150 – although Katal is the SI unity for enzymatic activity, please refer the xylanase activity also in units of activity (micromoles per minute) because it is most used and easier to compare with other equivalent enzymes.

- lines 160-163 – This paragraph is confused in describing the best strains in Figure 4 for xylanase activity. The second best strain is an Aspergillus strain!

- line 163 – Hypocrea sp. AB1XYL4 – it is named Trichoderma in Figure 4 !

- line 165 -  Hypocrea sp. BB2A1 – it is named Trichoderma in Figure 5 !

- Figure 7 – why Hypocrea lixii AB6XYL2 (Figure 4 and page 9, line 161) was not selected? It is referred in line 261 of Discussion as the second best fungi strain identified by authors for xylanase activity!

- line 266 – specify what kind of polymers

- line 278 – add the word “activity” after xylanolytic

- pages 16 and 17 – is it necessary to have 2 sessions for Authors Contribution?

Author Response

Please see the attachment with the comments of the reviewer 

Reviewer 2 Report

Comments and Suggestions for Authors

The manuscript is well written, however there are some missing parts.

1. Authors should check the abstract and revise it. It's better to make it more attractive. 

2. What is nkat? Readers might not know it as all people know Unit. And, why didn't give the enzmatic activity in Unit (U) instead of nkat? Further articles can easily compare their data with your paper. 

3. Are the isolated fungal strains (especially Aspergillus) known as GRAS or not? How we will use pathagonic fungi in industry?

4. How we can use isolated fungal strains and their enzymes in the industry? Is the enzymatic activity enough to degrade the lignocellulosic substrate? Authors should compare the data with commercial enzyme products which are commonly used in the industry. The authors avoided answering these questions. 

5. Xylanase is important to degrade lignocellulosic material, but it's not enough. We should have an enzyme cocktail. Is it economically visible to use only xylanase to degrade lignocellulose. 

Round 2

Reviewer 2 Report

Comments and Suggestions for Authors

No comments